# Importance of the X-ray edge singularity for the detection of relic neutrinos in the PTOLEMY project

Zhiyang Tan (谭志阳)[1,2] and Vadim Cheianov[1⋆]

**1** Lorentz Institute, Leiden University, Leiden, Niels Bohrweg 2, NL-2333 CA, The Netherlands
**2** Dahlem Center for Complex Quantum Systems and Physics Department,
Freie Universität Berlin, Arnimallee 14, 14195 Berlin, Germany

⋆ cheianov@lorentz.leidenuniv.nl

## Abstract

Direct detection of relic neutrinos in a beta-decay experiment is an ambitious goal that has long been beyond the reach of available technology. One of the most challenging practical difficulties for such an experiment is managing a large amount of radioactive material without compromising the energy resolution required to distinguish useful events from the substantial beta-decay background. The PTOLEMY project offers an innovative solution to this problem by depositing radioactive material on graphene. While this approach is expected to address the main challenge, it introduces new issues due to the proximity of the beta decayers to a solid-state system. In this work, we focus on the effect of the shakeup of the graphene electron system caused by a beta-decay event. We calculate the distortion of the relic neutrino peaks resulting from this shakeup, analyze the impact of the distortion on the visibility of neutrino capture events, and discuss potential technological solutions to enhance the visibility of these events.

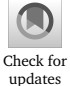

## 1   Introduction

The discovery of the cosmic microwave background (CMB) by Penzias and Wilson [1] was a pivotal event in Big Bang cosmology. The data from WMAP (Wilkinson Microwave Anisotropy Probe), 2001 [2] and Planck in 2013 [3] gave us the snapshot of the Universe dating back to about 13 billion years ago. However, the CMB provides no direct access to the Universe within 300 000 years of the Big Bang, because electromagnetic waves could not freely propagate in that epoch. For this reason, astronomers looked for an alternative messenger, a particle that had decoupled from matter earlier than the CMB. In particular, the decoupling of neutrinos is believed to have occurred about one second after the Big Bang. Unfortunately, the extremely weak coupling of neutrinos with matter makes the detection of the Cosmic Neutrino Background (C$\nu$B) a challenging task. Various ideas have been put forward as to how the C$\nu$B could manifest itself in a laboratory experiment [4–6]. The most practicable route to C$\nu$B detection today goes back to Weinberg's observation that the processes of cosmic neutrino capture should leave an extremely weak however potentially discernible feature in the beta spectra of radioactive nuclei [4]. Weinberg's original idea was elaborated in several proposals [7,8] centred around the beta decay of Tritium, which has a number of advantages such as the high neutrino capture cross-section, convenient half-lifetime, sufficient abundance and relatively simple chemistry [9].

Spontaneous beta-decay is a radioactive process in which a neutron decays into a proton, also emitting an electron and an anti-neutrino [10]. Its sibling process, induced beta decay occurs through the absorption of a neutrino. The two processes are illustrated in the following two reaction equations

$$^{3}\mathrm{H} \rightarrow\ ^{3}\mathrm{He}^{+} + e^{-} + \bar{\nu}_{e}\,, \tag{1}$$

and

$$\nu_{e} +^{3}\mathrm{H} \rightarrow\ ^{3}\mathrm{He}^{+} + e^{-}\,. \tag{2}$$

While the former is a $1 \rightarrow 3$ process resulting in a wide the continuous energy spectrum of beta-electrons, the latter is a $2 \rightarrow 2$ process imposing fixed values of the kinetic energies of the products in the centre of mass reference frame. For this reason, the energy spectrum of the beta-electron emitted in an induced process forms a narrow peak whose width is determined only by the variance of the kinetic energy of the incoming particles. Furthermore, considering the finite value of the neutrino mass $m_{\nu}$, this peak should be separated from the edge of the spontaneous beta-spectrum by the energy gap equal to $2m_{\nu}$.

The expected influence of the C$\nu$B on the beta emission of nuclei is illustrated in Fig. 1, which shows the theoretically predicted beta spectrum of free monoatomic $^{3}$H. The figure contains both the spontaneous beta decay background and the C$\nu$B contribution.[1] The two narrow C$\nu$B peaks in the spectrum come from the contributions of different neutrino mass eigenstates. The broadening of the peaks reflects the finite energy resolution of the experiment, which in this plot is assumed to be 40 meV [11, 12]. The energy gap is only about 100 meV, so in order for the C$\nu$B signal to be visible the energy resolution of the experiment has to be

---

[1]We acknowledge Dr. Boyarsky to provide us the date and the Mathematica code.

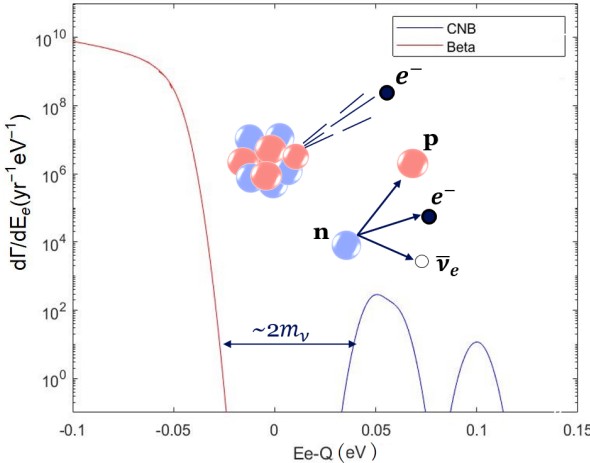

Figure 1: Beta decay spectrum and the sketch of the beta decay process. The blue curve depicts the C$\nu$B spectrum, and the red curve represents the beta decay spectrum, and they are separated by an energy gap, which has an order of magnitude twice the neutrino mass ($2m_\nu$). The annotation of the horizontal axis is the kinetic energy of the emitted electron measured from the endpoint of the beta decay spectrum, and the annotation of the vertical axis is the events observed at a certain energy per year. The area behind each cure represents the events per year, and the area behind the C$\nu$B curve is only 4.

on the same order of magnitude or less than the neutrino mass, otherwise the extremely weak cosmic neutrino feature will be submerged by the massive beta decay background.

Although it has been demonstrated that a 40 meV energy resolution of the beta-electron detector is within the reach of today's technology [13], the task of the fabrication of the neutrino target remains a serious challenge. The difficulty stems from the conflicting requirements of sufficiently high sample size on the one hand and sufficiently low probability of information loss to inelastic scattering on the other. In particular, detectors using gas-phase Tritium as a neutrino target, such as KATRIN, fall short of the required resolution [8] due to complications arising from inelastic collisions of electrons with ionised gas. Today, the most promising proposal addressing the issue is the solid state setup put forward by the PTOLEMY collaboration [13–15]. In this design, the required density of radioactive material will be achieved by creating a stack of two-dimensional graphene sheets covered by $^3$H. Inelastic collisions are avoided due to the emission of beta-electrons into the empty space between the graphene sheets, from where the electrons are guided into the calorimeter with the help of a cleverly designed electromagnetic guidance system.

While the PTOELMY design offers an appealing solution to one issue, it also encounters a range of challenges of its own. Recently, it was pointed out that trapping Tritium on a solid state surface leads to a substantial loss of energy resolution due to the zero-point motion of the nucleus [16]. Different mitigation strategies have been discussed such as engineering of the trapping potential [17], or use of heavier emitters [18–23], however, none of them is free of difficulty.

One of the important conclusions one can draw from these recent studies is as follows: the tightness of the energy resolution requirements imposed by the task of relic neutrino detection mandates exhaustive understanding and high control of the energy balance of beta-decay processes on a solid state surface down to the energy scale of about 10 meV. Clearly, the first step towards this goal should be the identification and preliminary theoretical analysis of all solid

state effects affecting the beta-electron on this energy scale. This, presumably, constitutes a considerable research program consisting of a range of tasks each addressing a separate solid state effect or interplay of several effects, if necessary. Some early and by no means comprehensive discussion of potentially harmful effects can be found in Refs. [11, 16, 17].

In this work, we aim to contribute to the theoretical analysis of solid state effects affecting the resolution of the PTOLEMY experiment by looking into a particular set of phenomena associated with the shakeup of the electron Fermi sea in graphene due to the beta-decay process. Ref. [11] identifies the Fermi sea shakeup as potentially very harmful due to the large bandwidth of graphene's conduction band. Our preliminary analysis here shows that the shakeup effect may not be as dangerous as it may seem from dimensional considerations. The main reason for this is that the shakeup of the Fermi sea results in a peculiar distortion of the line shape of a beta-electron, which is highly asymmetric and contains a power-law divergence at the energy of the process where no energy is transferred to the Fermi sea. Such a line shape is reminiscent of the famous X-ray edge singularity in the X-ray emission spectra (XES) of metals [24–29] and it is explained by the same physical mechanism. We demonstrate that the X-ray edge type broadening, although unpleasant, is not fatal for the PTOLEMY experiment, moreover, there are reasonably straightforward *in situ* ways to mitigate it. What we find a lot more dangerous is the effect of the core hole recombination. Our main conclusion here is that the requirements on the core hole lifetime are so stringent that one may need to consider ways to make the ion formed after the beta-decay chemically stable.

We would like to emphasize that in this work, we do not delve into the ramifications associated with the zero-point motion of the nucleus. The reader may assume that we are examining beta-decay within a system where such effects have been mitigated. This mitigation may be achieved through the utilization of heavy isotopes, such as $^{171}$Tm or $^{151}$Sm, or by implementing engineering strategies like confining potential manipulation, such as configurations where a Tritium atom is attached to a surface via physisorption. We defer the consideration of practicality related to the use of heavy isotopes or physisorption on graphene for a separate discussion.

## 2 Formulation of the problem

In this section, we introduce the spectral function, which encapsulates the influence of any solid-state environment on the shape of the beta-spectrum emitted by a decaying atom. Furthermore, we discuss some general properties of the spectral function, in particular, the effect of a finite lifetime of the decay product. We conclude this section by formulating a mathematical model of beta-emission on graphene in the presence of the Fermi sea of mobile electrons. We also discuss why despite being only an approximation to the intricate system under investigation, such a model captures the most important qualitative features of beta-decay, as well as providing reasonable quantitative estimates for the key parameters of the distortion of the beta-spectrum.

We consider a beta-emitter atom positioned at some microscopic distance from a graphene sheet. Possible mechanisms of attachment may include chemisorption or physisorption on graphene, adsorption on an atomically thin insulating layer grown on top of graphene, adsorption of a molecular coordination complex containing beta-emitter as a ligand, or other. In either case, we assume that the hybridisation between the orbitals of the beta-emitter and the electron system in graphene is perturbatively weak. To simplify the presentation, we focus on the capture process

$$\nu_e + {}^A_Z X^Q \rightarrow {}^A_{Z+1} X^{Q+1} + e^- . \tag{3}$$

Here $Q$ is the charge state of the beta-emitter and $Q + 1$ is the charge state of the daughter

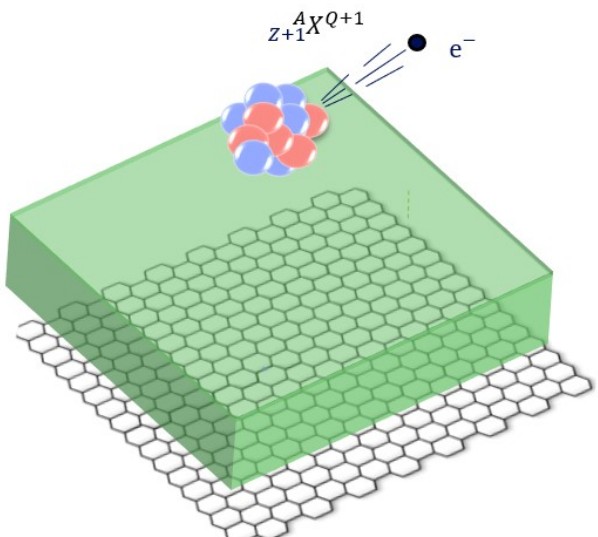

Figure 2: Sketch of the beta emitter and graphene sheet. $_{Z+1}^{A}X^{Q+1}$ is the production of the beta decay process, and it introduces a sudden localized perturbation in the electron system of graphene. This sudden localized perturbation causes an X-ray edge, which changes the beta decay spectrum. The thin green layer is a spacer inserted between the beta emitter and the graphene sheet to isolate the daughter isotope $_{Z+1}^{A}X^{Q+1}$ from the graphene sheet. Thus it improves the lifetime of $_{Z+1}^{A}X^{Q+1}$.

isotope. It is assumed that an electrically neutral atom corresponds to $Q = 0$. All general results discussed here extend to the case of spontaneous beta decay trivially. After capturing an incoming neutrino, the beta-emitter converts into a daughter isotope, releasing a fast outgoing electron as depicted in Fig(2). The sudden emergence of a charge centre next to the graphene sheet triggers local rearrangement of the material's electronic structure, in the exact same manner as the core hole in an X-ray emission experiment. In the latter case, such a rearrangement is known to result in the X-ray edge singularity in the emission spectrum [24–29], which is seen as the broadening of the emission line into an asymmetric shape having a power-law singularity at the emission edge. A similar phenomenon will occur in the $\beta$-spectrum of a radioactive atom on graphene.

Generally, the influence of the rearrangement of the electron system on the beta-spectrum is described by the following convolution (see section 2.2 for the technical details)

$$\frac{d\tilde{\Gamma}}{dE_k} = N\left(\frac{d\Gamma}{dE_k} * A\right)(E_k),\qquad(4)$$

where $d\Gamma/dE_k$ is the differential beta emission rate for a free monoatomic beta-decayer in the vacuum, $N$ is the number of beta-decayers and $A$ is the spectral function. It is worth noting that this expression is quite generic and it is not tied to any particular model of interaction of the beta-decayer with the solid state environment, as long as the electronic composition of the beta-decayer and the daughter ion can be defined in a meaningful way.

The spectral function encodes the internal dynamics of the solid state system after the sudden emergence of a positively charged daughter ion. Let $H_g$ denotes the Hamiltonian describing the graphene sheet in the presence of the charged daughter ion, and let $|\lambda\rangle$ be the eigenstates of this Hamiltonian

$$H_g|\lambda\rangle = E_\lambda|\lambda\rangle,\qquad(5)$$

then the spectral function is given by the following expression (see section 2.2)

$$A(E) = \sum_{\lambda} |\langle \lambda | FS \rangle|^2 \, \delta(E + E_\lambda - E_0) \,. \tag{6}$$

Here $E_0$ is the ground state energy of the graphene-beta emitter system prior to beta-decay, and $|FS\rangle$ is the ground state of the solid-state system prior to beta decay. Implicit within Eq. (6) is the assumption that the decay process transpires significantly faster than the typical timescales associated with the dynamics governed by $H_g$. The derivation of Eq. (6) from Fermi's Golden Rule is contained in 2.2.

## 2.1 Properties of the spectral function

From Eq. (6) one can infer the following general properties of the spectral function

- Non-negativity

$$A(E) \geq 0 \,.$$

- Normalisation

$$\int_{-\infty}^{\infty} dE A(E) = 1 \,.$$

- End point of the support

$$A(E) = 0 \,, \qquad E < E_0 - E_{\text{GS}} \,.$$

Here $E_{\text{GS}}$ is the energy of the ground state of the Hamiltonian $H_g$.

Most of this work is going to be focused on the properties of the function $A(E)$ near the end point of its support. However, before delving into this discussion an important remark is in order here.

If the interaction between the daughter ion and its solid state environment were negligible, then the ion would have been an eigenstate of $H_g$ and $A(E)$ would have had the structure of a delta-peak $A(E) = \delta(E + E_i)$. In reality, the ion gets entangled with the environment through different mechanisms, which leads to the peak's broadening. One mechanism, which deserves particular attention is the hole capture. It works as follows. Due to the work function difference, an atom attached to graphene would typically donate or accept a number of electrons. This effect is characterised by the atom's equilibrium charge state, that is the atomic number less the average total number of electrons occupying its atomic shells in equilibrium. For an mother isotope $^A_Z X$ in the charge state $Q$ the dominant decay channel will be into a daughter isotope $^A_{Z+1} X$ in the charge state $Q + 1$, as is described by Eq.(3). Depending on the chemistry of the graphene-atom interaction $^A_{Z+1} X^{Q+1}$ may or may not be an equilibrium charge state. If this state is not equilibrium, it will further decay into an equilibrium charge state through, for example, an electron capture, which in the language of elementary excitations of the solid state system amounts to the creation of a hole $h^+$. The correct equation describing such a process would be

$$\nu_e + {}^A_Z X^Q \rightarrow {}^A_{Z+1} X^Q + e^- + h^+ \,, \tag{7}$$

in which the hole $h^+$ is a quasiparticle endowed with its own energy and momentum. As a $2 \rightarrow 3$ process, Eq. (7) would not result in a sharp relic neutrino peak in the beta spectrum, rather it would produce a broad continuum useless for relic neutrino detection.

The problem can be avoided if the charge state $^A_{Z+1} X^{Q+1}$ formed immediately after beta-decay is stable, or, at least, long-lived. In that case, the spectral function $A(E)$ will have a sharp resonant peak at the energy corresponding to the $2 \rightarrow 2$ process Eq.(3), while the process Eq.(7) will be present at the tail. In such a limit, one can treat the metastable daughter

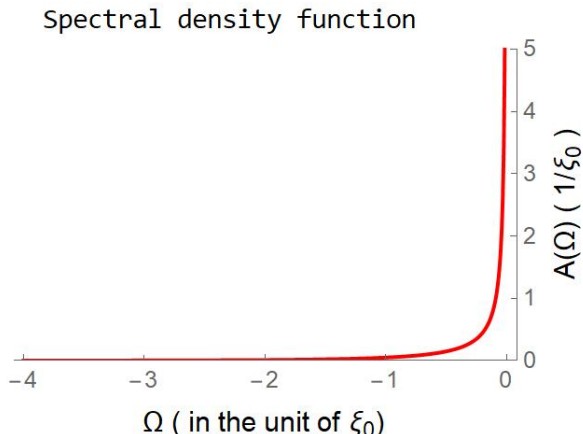

Figure 3: The spectral density function of graphene at a given energy. $A(\Omega)$ describes the possibility of graphene absorbing the outgoing electron's energy. The energy is measured in the unit of cut-off energy $\xi_0$.

ion $_{Z+1}^{A}X^{Q+1}$ as the final product of the decay process, however allowing for Heisenberg's uncertainty in the energy conservation law. More precisely, let $\tau$ be the lifetime of the daughter ion $_{Z+1}^{A}X^{Q+1}$. By virtue of Heisenberg's principle, this will introduce an additional $\hbar/\tau$ uncertainty into the energy conservation law for the products of the decay and consequently into the energy of the beta-electron. It is tempting to think that in order for the neutrino peak to be visible such an uncertainty has to be less than the neutrino mass, $\tau m_\nu c^2/\hbar \ll 1$. In fact, the situation is more complex due to the effect of a spillover of the massive beta decay background into the vicinity of the relic neutrino peak. As will be discussed later in this work, in order to suppress the spillover effect and make the neutrino capture peak observable, the lifetime of $_{Z+1}^{A}X^{Q+1}$ has to be extremely long. For example, in the case of beta decay of neutral Tritium, the lifetime of the $^3\text{He}^+$ daughter ion needs to be longer than $10^{-2}$ s. Such long lifetimes require that either the atom be well insulated from graphene, for example by a thin layer of a wide gap insulator, or $_{Z+1}^{A}X^{Q+1}$ be an equilibrium charge state. The discussion of strategies for making $_{Z+1}^{A}X^{Q+1}$ a long-lived state is outside the scope of the present work and it will be addressed elsewhere.

The rest of our analysis is devoted to situations where the conditions on the lifetime of the ion are fulfilled so we may assume that the ion is for all intents and purposes stable. In this case, the dominant mechanism of entanglement of the ion with the quantum degrees of freedom of graphene is through the Coulomb interaction between the suddenly emerging localised charge of the daughter ion and graphene's electron system. The comprehensive analysis of such a sudden quench is, of course, a daunting, if at all feasible task. However, as we shall establish later in this paper, the spectral function possesses a universal structure near the edge of its support, which makes the detailed knowledge of the rest of the energy spectrum unnecessary. More precisely, the spectral function has the shape of a very sharp one-sided peak (see Fig. 3) with a power-law divergence at the edge, which ensures the visibility of the relic neutrino feature. Moreover, due to the one-sidedness of the peak, there is no spillover effect from the beta-decay background. The analysis of the shape of the peak only requires the knowledge of the effective low-energy theory of the ion-graphene interaction, which massively simplifies theoretical analysis of the process.

## 2.2 Considerations of the model

We can now state the main assumptions about the system that will enable us to focus on the Fermi sea shakeup effect

(1) The recoil of the daughter ion from the beta decay will be neglected. This effect and its mitigation strategies have been addressed in recent literature [16, 17, 23, 30]. Although significant on the 10 meV scale, this effect is not directly related to the Fermi sea shakeup discussed here.

(2) We only focus on the processes that do not result in the excitation of internal degrees of freedom of the daughter ion. Only such processes are relevant to the neutrino capture experiment.

(3) We assume that the graphene sample is spatially uniform. Where we analyse the effect of disorder, we assume that the disorder is spatially uniform.

(4) We neglect crystalline lattice effects arising from the lateral position of the beta-decayer relative to the graphene sheet. This is because the X-ray edge singularity is an infrared phenomenon, arising from the long-range Coulomb interaction, which is insensitive to the exact position of the Coulomb centre inside the unit cell.

(5) We assume for simplicity that the initial charge state of the beta-decayer is neutral.

(6) We assume that the temperature of the system is less than the resolution of the detector, which is about 100K. We thus neglect thermal effects and consider our model at zero temperature.

Based on the above assumptions, we formulate the Hamiltonian

$$H = H_0 + H_w + H_{D-G}, \tag{8}$$

which comprises the kinetic part, the weak interaction part, and the coupling between the daughter ion and graphene. The kinetic part is trivial, and it is the total relativistic energy of free motion of all particles involved in the process

$$H_0 = H_\nu + H_D + H_M + H_e + \sum_s \sum_k s\hbar v_F k C_{ks}^\dagger C_{k's}. \tag{9}$$

Here $H_\nu$ stands for the kinetic energy of the neutrino, $H_e$ – for the kinetic energy of the beta-electron, $H_M$ and $H_D$ are the Hamiltonians of the isolated mother and the daughter isotopes respectively, $v_F$ is the Fermi velocity $C_{ks}^\dagger$ and $C_{ks}$ are the creation and annihilation operators for the electrons in the graphene respectively, and s is the band index of the graphene. The spin and valley indices have been absorbed into the band index.

The term

$$H_w = \int G_F \nu M D^\dagger e^\dagger d^3x, \tag{10}$$

is the effective Hamiltonian of the beta-decay process, where $\nu^\dagger$ is the creation operator for the neutrino, $D^\dagger$ is the creation operator for the daughter ion, and $M$ is the annihilation operator for the mother isotope atom, $e^\dagger$ is the creation operator for the outgoing electron, and $G_F$ is the effective beta decay interaction constant absorbing the details of ultrafast electroweak processes inside the nucleus [10, 31],

The final term in the Hamiltonian, $H_{D-G}$ describes the interaction between the electrons in

graphene and the charged daughter ion, which suddenly emerges due to the beta-decay process. Since the daughter ion is very heavy and therefore can be treated as a localised object,[2] we can write its density operator as $D_0^\dagger D_0 \delta(\mathbf{r})$, where the origin $\mathbf{r} = 0$ coincides with the equilibrium position of the mother atom and $D_0^\dagger D_0$ is the daughter ion counting operator. Thus we write this interaction term as follows

$$H_{D-G} = -\int \rho(x)V(x)D_0^\dagger D_0 d^2 x \,, \tag{11}$$

where

$$V(x) = \frac{e^2}{4\pi\epsilon\epsilon_0 \sqrt{d^2 + x^2}} \,, \tag{12}$$

is the Coulomb pontential of the ion in the graphene plane, and

$$\rho(x) =: \psi^\dagger(x)\psi(x): \,, \tag{13}$$

is the two-dimensional density of electrons in graphene. We have denoted the dielectric constant of the graphene as $\epsilon$ and the distance between the daughter ion and the graphene as $d$. The field operator of an electron in graphene $\psi(x)$ admits for the standard plane wave decomposition

$$\psi(x) = \sum_{k,s} C_{ks}\psi_{ks}e^{-i\vec{k}\vec{x}} \,, \tag{14}$$

where $\psi_{ks}$ is the spinor of the Dirac electrons in the graphene,

$$\psi_{ks} = \frac{1}{\sqrt{2}}\begin{pmatrix} e^{-i\theta_k} \\ s \end{pmatrix} \,, \tag{15}$$

with $\theta_k = \arctan(k_y/k_x)$.

In the standard vein of beta decay theory, the transition probability is given by Fermi's golden rule

$$W = \frac{2\pi}{\hbar}\sum_f \left| \langle 0|_M \langle 0|_\nu \langle k|_e \langle 1|_D \langle\lambda| \int G_F \nu M D^\dagger e^\dagger d^3 x |1\rangle_M |1\rangle_\nu |0\rangle_e |0\rangle_D |FS\rangle \right|^2 \delta(E_f - E_i), \tag{16}$$

where $f$ refers to the final states of the process, and $|\lambda\rangle$ is the final eigenstate of the graphene Hamiltonian with the energy $E_\lambda$ and $|FS\rangle$ is the Fermi sea of the graphene electrons characterised by the ground state energy $E_0$, $|0\rangle_{M,D}$ denotes a state where the isotope (mother or daughter) is absent, and $|1\rangle_{M,D}$ denotes a state where the isotope is present in the state of zero kinetic energy. The equation neglects the effect of the recoil of the daughter isotope. Such an effect has been extensively analysed elsewhere [16], and it leads to additional broadening of the neutrino capture peak.

The total energy of the final state in the process is $m_e c^2 + \frac{\hbar^2 k^2}{2m_e} + E_\lambda + m_D c^2$, and the energy of the initial state is $m_\nu c^2 + m_M c^2 + E_0$. The final state is the tensor product of the neutrino state, the outgoing electron state, and the graphene electron state. We can sum over the normal beta decay part and graphene part independently. We use index $f'$ to denote the final state of beta decay and use index $\lambda$ to denote the final state of electrons in graphene.

---

[2]In fact, the departure from the local approximation due to the quantum zero-point motion of the beta-decayer is another source of the uncertainty of the emitted electron energy [16]. It could, in principle, be suppressed by choosing heavier beta-decayers, e.g. rare earth atoms. We do not address the zero-point motion effect in this work assuming that it is less important than the shakeup effect analysed here.

It is worth noting that Eq. (16) reflects the fact that following the beta-decay of the nucleus, which we treat as an instantaneous process, the combined graphene-electron system emerges in a quantum superposition of states $|f\rangle = \sum_{k,\lambda} C_{k,\lambda} |k\rangle_e |\lambda\rangle$. The states in the superposition have to obey the energy conservation law $(\hbar^2 k^2/2m_e) + E_\lambda = \text{const.}$ Therefore, the tensor $C_{k,\lambda}$ is inseparable and the final state is an entangled state between the outgoing electron and the graphene system in which the kinetic energy of the beta electron is indefinite. Therefore despite their dynamics being decoupled, their entanglement necessitates the total transition probability to be a convolution of the transition probability function of beta decay and the graphene spectral density function, as depicted by the subsequent expression.

$$W = \frac{2\pi}{\hbar} \int dE \sum_{f'} \sum_{\lambda} |V_{if'}|^2 |\langle \lambda | FS \rangle|^2 \delta(E + E_\lambda - E_0) \times \delta(E_k - Q - E) = (\Gamma * A)(E_k), \quad (17)$$

where $V_{if'}$ is the weak interaction matrix element, $Q = m_M c^2 + m_\nu c^2 - m_e c^2 - m_D c^2$ is the emission energy of beta decay, $E_k$ is the kinetic energy of the beta electron, $\Gamma(E)$ is the beta decay transition rate of a single atom in the vacuum, and $A(E)$ is the electron spectral density function in graphene. Note, that the energy conservation law in Eq. (17) neglects the kinetic energy of both the heavy particles and the neutrino. The latter is due to the fact that we are interested in the narrow vicinity of the endpoint of the beta decay spectrum, furthermore, the temperature of the cosmic neutrino background, 1.95 K, is negligible compared to the neutrino's mass. Therefore, Eq. (17) both applies to the beta decay background and the cosmic neutrino absorption process. Following the convention, we can consider $d\tilde{\Gamma}/dE_k$ as observable beta decay rate at given electron kinetic energy corrected by the electron shakeup process

$$\frac{d\tilde{\Gamma}}{dE_k} = N \frac{dW}{dE_k} = N \left( \frac{d\Gamma}{dE_k} * A \right)(E_k), \quad (18)$$

where $N$ is the number of beta-decayers deposited on graphene. The discussion above is also valid for the process of background neutron decay. We have found out that the smeared beta decay spectrum is nothing but the convolution between the spectral density function of graphene and the original beta decay spectrum. This result is vital to us since it tells us the influence of the X-ray edge in the PTOLEMY project. Later, we will show that the spectral density function does have an X-ray edge.

## 3 Linked cluster expansion

In this section, we assume that the daughter ion is screened instantaneously. This assumption does not work well in graphene and we will revise it later, however, at this stage, we would like to keep our discussion as simple as possible focusing on the key physics of the problem. With the help of the RPA and neglecting time retardation, we obtain the effective static dielectric constant of graphene [32]

$$\epsilon = \frac{1 + \kappa}{2} + \frac{2\pi e^2}{q} \frac{g_s g_v q}{16 \hbar v_F}, \quad (19)$$

where $\kappa$ is the external dielectric constant of the substrate, the degeneracy factor $g_s = 2$, $g_v = 2$, and $v_F \approx 10^6$ m/s. If the graphene is suspended in the vacuum, then $\epsilon \approx 4.4$.

The spectral density function $A(E)$ is the central object of interest, so we will investigate

its property further in this section. The expression for it is

$$
\begin{aligned}
A(E) &= \sum_{\lambda} |\langle \lambda | FS \rangle|^2 \, \delta(E + E_\lambda - E_0) \\
&= \frac{1}{2\pi\hbar} \sum_{\lambda} \int dt \, \langle FS | e^{-iH_g t} | \lambda \rangle \langle \lambda | FS \rangle \, e^{-i(E-E_0)t/\hbar} \\
&= \frac{1}{2\pi\hbar} \int dt \, \langle FS | e^{-iH_g t/\hbar} | FS \rangle \, e^{iE_0 t/\hbar} e^{-iEt/\hbar} ,
\end{aligned}
\tag{20}
$$

where $H_g$ is the Hamiltonian of the graphene after the beta decay. The influence of the energy redistribution between the graphene system and the daughter ion on the energy spectrum is encapsulated within the spectral density function, defined by Eq. (20). We express the Hamiltonian $H_g$ in momentum space [32,33],

$$
H_g = \sum_s \sum_k s\hbar v_F k C_{ks}^\dagger C_{k's} - \frac{1}{L^2} \sum_{ss'} \sum_{kk'} V(k, k') F_{ss'}(k, k') C_{ks}^\dagger C_{k's'} ,
\tag{21}
$$

with

$$
F_{ss'}(k, k') = \frac{1}{2} \left[ ss' + \exp(i\theta_k - i\theta_{k'}) \right] ,
\tag{22}
$$

and

$$
V(k, k') = \frac{2\pi e^2 \exp\left(-|k' - k|d\right)}{\epsilon |k - k'|} .
\tag{23}
$$

To calculate the spectral density function $A(E)$, we first focus on the density function, which is its Fourier transform

$$
\rho(t) = \langle FS | e^{-iH_g t/\hbar} | FS \rangle \, e^{iE_0 t/\hbar} .
\tag{24}
$$

One can recognize $-i\Theta(t)\rho(t)$ as the core hole Green function in the conventional X-ray singularity problem [25, 26, 34–36]. Using linked cluster expansion method [36–38], we can express the density function $\rho(t)$ in another way.

$$
\rho(t) = \exp\left[ \sum_l F_l(t) \right] ,
\tag{25}
$$

where $F_l(t)$ is the l-th connected diagrams,

$$
F_l(t) = \frac{(-i)^l}{l} \int_0^t dt_1 \cdots \int_0^t dt_l \, \langle TV(t_1) \cdots V(t_l) \rangle_{\text{connected}} .
\tag{26}
$$

$V(t)$ is the interaction term in the Dirac picture. We note that $F_1(t) = -iE_i t$ where the constant $E_i$ is called the self-energy. The self-energy is responsible for the overall energy shift of the beta spectrum relative to the vacuum one. Assuming the interaction is weak, we can restrict ourselves to the second term [36].

$$
F_2(t) = -\int_0^\infty \frac{du}{u^2} R_e(u)(1 - e^{-iut}) ,
\tag{27}
$$

where

$$
R_e(u) = \frac{1}{\pi L^2} \sum_q |V(q)|^2 \Lambda(q, u) ,
\tag{28}
$$

and

$$\Lambda(q,u) = \text{Im}P^{(1)}(q,u),\tag{29}$$

where $P^{(1)}(q,u)$ is the polarizability function. The expression of the polarizability is given in many references [36, 39], so we just quote the result

$$P^{(1)}(q,u) = \frac{-ig'}{L^2}\int_{-\infty}^{\infty}\frac{dE}{2\pi}\sum_{p,s,s'}|F_{ss'}(p,p+q)|^2 \times G_{ss}(p,E)G_{s's'}(p+q,E+u),\tag{30}$$

where $G_{ss}(P,E)$ is the Green function for graphene, $g'$ is the degeneracy, and it is 4 for intrinsic graphene. The polarizability function of the intrinsic graphene was calculated in the references [32, 40, 41]. In particular, its imaginary part gives

$$\Lambda(q,u) = \frac{q^2}{4\sqrt{u^2 - v_F^2 q^2}}\Theta(u - v_F q).\tag{31}$$

To calculate $R_e(u)$, one needs to substitute Eq. (31) into Eq. (28). To simplify the calculation, we omit the exponential factor in the Eq. (23), and the potential becomes

$$V(k,k') = \frac{2\pi e^2}{\epsilon|k-k'|}.\tag{32}$$

Although this crude simplification is only valid near the edge of the spectral density function, it gives us a physics insight in the first step. Later, we will recover the effect of the distance d in the next section. Then we have

$$R_e(u) = gu,\tag{33}$$

where

$$g = \frac{e^4}{2\epsilon^2 v_F^2}.\tag{34}$$

Using the effective dielectric constant we get in the previous step, we can find $g \approx 0.125$. Substituting it into the expression of $F_2$, thus one can get

$$F_2(t) = -g\int_0^{\xi_0}\frac{(1-e^{-iut})du}{u} \approx -g\int_{1/it}^{\xi_0}\frac{du}{u} \approx -g\ln(1+it\xi_0).\tag{35}$$

Note that due to the logarithmic divergence of the integral we had to use an arbitrary ultraviolet cutoff $\xi_0$ to complete the calculation. This cutoff parameter is important for the estimate of the visibility of the C$\nu$B peak, therefore it cannot be chosen arbitrarily. The physical meaning and the value of $\xi_0$ will be clarified later.

The expression (35) is valid only for large enough $t$, that is $\xi_0 t \gg 1$. The expression for the density function is obtained from Eq. (35),

$$\rho(t) = \exp(-iE_i t - g\ln(1+it\xi_0)),\tag{36}$$

where the self-energy term $E_i$ can be absorbed into the exponent $E$ in the Fourier transformation in the later steps. It has no effect on the shape of the spectral density function, but shifts it by $E_i$.

# 4 Spectral density function

In this section, we give an explicit formula for the spectral density function of graphene. Furthermore, we investigate the influence of the height from the daughter ion above the graphene sheet, as well as the influence of disorder and the dynamical screening on the spectral density function.

The Fourier transform of Eq. (36) gives the spectral density function of graphene at the given energy, and it is exactly the gamma distribution function.

$$A(\Omega) = \int_{-\infty}^{\infty} \frac{dt}{2\pi} e^{-iEt} \rho(t) = \Theta(-\Omega) \frac{\exp(\Omega)}{\xi_0 \Gamma(g)(-\Omega)^{(1-g)}}, \tag{37}$$

with

$$\Omega = (E + E_i)/\xi_0. \tag{38}$$

The spectral density function obeys the gamma distribution, and the result coincides with that for a metal with a local constant interaction [36–38]. The gamma distribution function has two parameters: one is the shape parameter, which is the coupling constant in our problem, and another one is the scale parameter, which has so far remained an arbitrary cutoff parameter. The coupling constant determines how sharp the peak is near the edge, and the cutoff energy determines how many events are lost in the long tail of the spectral density function. Fig. 3 represents the spectral density function. As we can see in Fig. 3, the function sharply diverges when $\Omega$ goes to zero. In other words, the differential emission rate is the largest when no electron excitations are created in graphene at the end of the process.

## 4.1 Influence of the height

In the previous discussion, we calculated the spectral density function and obtained its shape parameter $g$. However, the cutoff energy remains unknown. Typically, one would expect the cutoff energy to be associated with the electron bandwidth or the Fermi energy. In this section, we find that the cutoff energy is dictated by the height of the daughter ion above the graphene sheet. Indeed, for any given $d$ the expression of $R_e(u)$ is

$$R_e(u) = e^4 \int_0^{u/v_F} \frac{qe^{-2qd}dq}{2\epsilon^2 \sqrt{u^2 - v_F^2 q^2}} = gu + \frac{\pi g u}{2} \left[ L_1 \left( \frac{2du}{v_F} \right) - I_1 \left( \frac{2du}{v_F} \right) \right], \tag{39}$$

where $L_1 \left( \frac{2du}{v_F} \right)$ and $I_1 \left( \frac{2du}{v_F} \right)$ are modified Struve function and modified Bessel function, respectively.
This equation may look slightly intimidating however its intuitive meaning is simple. The X-ray edge singularity is an infrared phenomenon, so only long-distance interaction is significant for it. Thus we can eliminate the short-distance structure of the Coulomb interaction. If $q \ll 1/(2d)$, the Yukawa type potential in Eq. (39) recovers to the normal Coulomb potential, thus giving a natural cutoff energy $\xi = \hbar v_F / 2d$. From asymptotic analysis (details are in Appendix 1), we find that the spectral density function was the same form as in Eq. (37) albeit with a cutoff energy $\xi'$,

$$\xi' = \frac{\xi_0}{1 + 2d\xi_0/v_F}. \tag{40}$$

$\xi_0$ is a microscopic parameter on the order of the bandwidth energy. Assuming that $\xi_0 \gg v_F/d$, one finds

$$\xi = \lim_{\xi_0 \to \infty} \xi' = \frac{\hbar v_F}{2d}. \tag{41}$$

The expression (41) should replace $\xi_0$ as the scale factor in the line shape equation Eq. (37). One can see that the scale factor depends on the height $d$, so one could potentially manipulate the line shape by adjusting the height of the ion.

## 4.2 Influence of disorder

Until now, we have only considered intrinsic graphene, uncontaminated and ungated. However, a finite density of beta emitters in graphene inevitably introduces disorder, which changes the mean free path and the density-density response function of electrons. Intuitively, one should expect that if the time scale associated with the inverse energy resolution of the beta detector is greater than the mean free time in graphene, i.e., $\tau_{\mathrm{res}} \gg \tau$, then one needs to take into consideration the effect of electron diffusion on the polarizability function and thus the coupling constant.

We consider Eq. (20), in the presence of many random impurities, focusing on the impurity-average effect. The impurity-average polarizability function $P_{\mathrm{imp}}(q, \omega)$ is [39, 42, 43]

$$P_{\mathrm{imp}}(q, \omega) = \frac{P^{(1)}(q, \omega + \frac{i}{\tau})}{1 + (1 - i\omega\tau)^{-1}\left[\frac{P^{(1)}(q, \omega + \frac{i}{\tau})}{P^{(1)}(q, 0)} - 1\right]}, \tag{42}$$

where $\tau$ is the momentum relaxation time of a quasiparticle. The required energy resolution of the detector is $\omega_{\mathrm{res}}$ is about 10 meV, therefore the resolution time $\tau_{\mathrm{res}}$ is $\sim 10^{-13}$ s. If $\tau > \tau_{\mathrm{res}}$, then one can neglect the effect of impurities, and the polarizability function recovers to the Lindhard function [42]. Otherwise, one needs to consider both diffusive and ballistic regimes.

To get some insight into the degree of coverage at which it is necessary to take the impurity scattering into consideration, we give an elementary estimate based on the mean free time calculated within the midgap model which is known to work reasonably well for hydrogen and other atoms with covalent on-site bonding on graphene [44–46]. In this model, mobility electrons in graphene see a carbon site with attached hydrogen as a vacancy. The potential is profiled as

$$U(r) = \begin{cases} \infty, & \text{if } 0 < r \le R', \\ U_0, & \text{if } R' < r \le R, \\ 0, & \text{if } r > R, \end{cases} \tag{43}$$

where $R'$ is the the radius of a hydrogen atom, and $R$ is the radius of a carbon atom. The mean free time is defined by the following expression

$$\frac{\hbar}{\tau_k} = \frac{8n_i}{\pi\rho(E_k)}\sin^2\delta(k), \tag{44}$$

where $\delta(k)$ is the phase shift of electrons with momentum k, and $\rho(E_k)$ is the density of state, with the explicit form

$$\rho(E_k) = \frac{2E_k}{\pi\hbar^2 v_F^2}. \tag{45}$$

Using the method in the paper [45], we can obtain the phase shift at $k \approx 0$,

$$\delta(k) = -\frac{\pi}{2}\frac{1}{\ln(2kR')}. \tag{46}$$

We expand the sine function in Eq. (44) for $kR' \ll 1$. Thus it gives the mean free time

$$\tau_k = \frac{\hbar\rho(E_k)}{2\pi n_i}(\ln kR')^2 = \frac{k}{\pi^2 v_F n_i}(\ln kR')^2, \tag{47}$$

where $n_i$ is the impurity concentration. The mean free path has a minimal value at $kR' = e^{-2}$, and its minimum value is

$$\tau_{min} = \frac{4e^{-2}}{\pi^2 v_F n_i R'} \, . \tag{48}$$

If $\tau_{min}$ is much greater than the resolution time, we can neglect the impurity effect. It means that

$$\frac{4e^{-2}}{\pi^2 v_F n_i R'} \gg 1/\omega \sim 10^{-13} \, s \, . \tag{49}$$

$R'$ is about 1 Å, so we find the condition for a maximum impurity concentration,

$$n_i \ll \frac{4e^{-2}\omega}{\pi^2 v_F R'} = 5 \times 10^{11}/\text{cm}^2 \, . \tag{50}$$

The mass density for the impurity in the case of hydrogen atoms is

$$\rho_i \approx 8.31 \times 10^{-13} \, \text{g}/\text{cm}^2, \tag{51}$$

and the standard surface density of graphene is about $7.61 \times 10^{-8}\,\text{g}/\text{cm}^2$ [47]. To conclude, for atoms in the onsite boding configuration the effect of disorder is negligible for the surface coverage of less than $5 \times 10^{11}\,\text{cm}^{-2}$. For atoms in other configurations the critical concentration could be different, even though we do not expect the difference to be dramatic. If impurity atoms are separated from graphene by a thin layer of insulator the critical coverage may be significantly greater.

Apart from impurities, phonons can also influence the mean free time of graphene. However, one can keep the system at a very low temperature to reduce the phonon scattering. For intrinsic graphene under liquid nitrogen temperature, the momentum relaxation time is about one ps [48, 49], which is one order of magnitude greater than the resolution time. Therefore, we can treat electrons of the graphene ballistically within the resolution time.

### 4.3 Influence of dynamic screening

The instantaneous screening assumption is not valid in the tail of the spectral function of graphene. While, if the weight of the tail is too high, it will cause the beta decay spectrum to broaden widely, and it may take many years to observe a single event. To get a more rigorous result, we need to consider the dynamic screening effect. The dynamic screening potential is [36]

$$V(q, \omega) = \frac{V_i(q, \omega)}{\epsilon(q, \omega)} \, . \tag{52}$$

For a sudden external potential [50]

$$V_i(q, \omega) = \frac{V(q)i}{\omega + i\delta} \, . \tag{53}$$

$\delta$ is an infinitesimal quantity, and $V(q) = 2\pi e^2/(\kappa q)$ is the bare Coulomb potential with $\kappa$ the external dielectric constant of the substrate. We notice that $\epsilon^{-1}(q, \omega) - 1$ is the susceptibility function so we can use the Kramers-Kronig relations.

$$\frac{1}{\epsilon(q, \omega)} - 1 = -\frac{1}{\pi} \int_{-\infty}^{\infty} d\omega' \text{Im}\left(\frac{1}{\epsilon(q, \omega')} - 1\right) \frac{1}{\omega - \omega' + i\delta} \, . \tag{54}$$

Substituting it into the expression of dynamic screening potential and performing the Fourier transform, one can find the time-dependent screening potential.

$$
\begin{aligned}
V(q,t) &= \int_{-\infty}^{\infty} V(q,\omega)e^{-i\omega t}d\omega \\
&= \frac{V(q)\theta(t)}{\pi}\int_{-\infty}^{\infty}\frac{1-e^{-i\omega' t}}{\omega'}\text{Im}\left(\frac{1}{\epsilon(q,\omega')}-1\right)d\omega' + V(q)\theta(t).
\end{aligned}
\tag{55}
$$

In the large time limit, the oscillating term disappears, so the potential becomes

$$
V(q,\infty) = \frac{V(q)}{\epsilon(q,0)}.
\tag{56}
$$

For graphene, $\epsilon(q,0)$ does not depend on q, so we can denote $\epsilon(q,0)$ as $\epsilon$. The first term in the last line vanishes in the short time limit, so the potential turns into the bare potential.

$$
V(q,0) = V(q).
\tag{57}
$$

For an arbitrary time, we can express $V(q,t)$ as

$$
V(q,t) = \frac{V(q)\theta(t)}{\epsilon(q,0)} - \frac{V(q)\theta(t)}{\pi}\int_{-\infty}^{\infty}d\omega'\frac{e^{-i\omega' t}}{\omega'}\times\text{Im}\left(\frac{1}{\epsilon(q,\omega')}-1\right).
\tag{58}
$$

From detailed analysis (see B), we found out that the spectral density function is the same as the previous result in Eq. (37), but with a different coupling constant

$$
g_1 = g + \int_{1}^{\infty}\frac{2g\epsilon}{\pi}\frac{a\log(\omega+\sqrt{\omega^2-1})}{\omega^2(\omega^2-1+a^2)}d\omega,
\tag{59}
$$

where $a = (\epsilon - 1) = \pi e^2/2\hbar\kappa v_F$, and this result is quoted from the Appendix 2 Eq. (B.6 ). It is similar in meaning to the fine structure constant in QED.

If graphene is suspended in the vacuum, then $g = 0.125$, and $g_1 \approx 0.200$. For graphene on $SiO_2$, the coupling constant g is about 0.065, and $g_1 \approx 0.090$.[3] As one can see in Fig. 4, the dynamic screening effects of intrinsic graphene significantly change the coupling constant at a small dielectric constant, and this can be understood intuitively. For intrinsic graphene, the Fermi energy is zero, therefore there is no intrinsic time scale. For long wave-length scattering, the screening time is about $1/(v_F q)$, which is also the time scale for the Fermi sea shakeup effect at the energy scale $v_F q$, so we cannot separate the screening of the Coulomb potential from the formation of the X-ray edge singularity. Hence, the dynamical screening effects have a considerable influence on the coupling constant. In contrast, the Fermi energy is non-zero for normal metals and gated or doped graphene, so the static screening is a good approximation near the edge in those cases. In the large dielectric constant limit, one can see from Fig. 4b, that the ratio of the dynamic coupling constant and static coupling constant approaches 1. It can be understood from Eq. (19) and Eq. (59) that when the external dielectric constant is big enough, it makes the dominant contribution to the total dielectric constant.

To summarize, the dynamical screening effect has a considerable influence on the coupling constant in intrinsic graphene, but it can be suppressed by applying an external gate voltage or increasing the dielectric constant of the substrate.

---

[3]The coupling constant $g_1$ is calculated from Eq. (59), and the coupling constant g is from the Eq. (34).

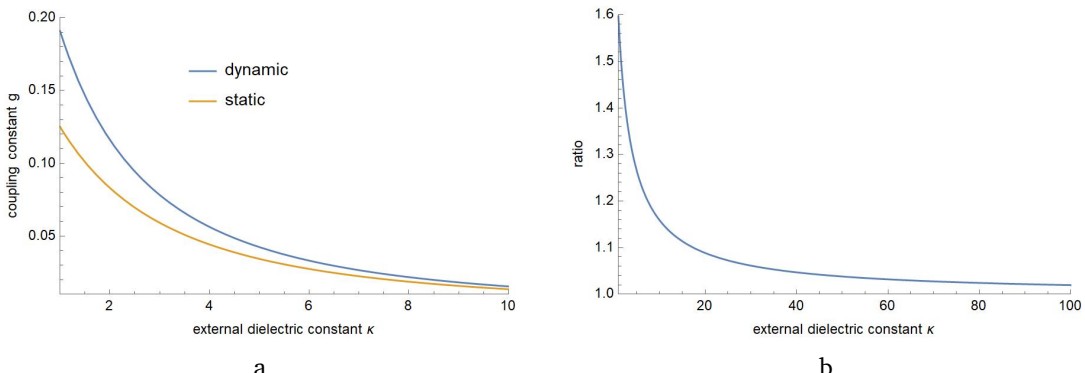

Figure 4: Fig. 4a depicts the coupling constant with different external dielectric constants. $\kappa$ describes the dielectric constant of the substrate, and g is the graphene's coupling constant. The blue curve represents the coupling constant considering the dynamic screening effect, and the orange curve represents the coupling constant in the static screening approximation. Fig. 4b describes the ratio of the dynamic coupling constant and the static coupling constant. In the large dielectric constant limit, the ratio approaches 1.

## 4.4 Constraints on the daughter ion's lifetime

In the previous discussion, we assumed that the lifetime of the daughter ion is infinite, however in reality this may not be true. The daughter ion could be chemically unstable and have 32propensity to, for example, capture an electron from its chemical environment. By virtue of Heisenberg's principle, shorter lifetimes would introduce greater uncertainty into the measured energy of the beta electron. In the conventional X-ray edge singularity context the lifetime of an ion is limited by the process of recombination of a conductance electron with the core hole. Such a process requires accommodation of a large amount of energy which is usually achieved through the Auger effect. In the case of an ion which is formed through beta decay, the electron deficit occurs in the chemically active outer shell therefore capture of an electron may be possible through a direct tunnelling process. The efficiency of such a process will vary depending on the way an atom is deposited on the surface. Insertion of a dielectric spacer between the atom and a graphene layer, or deposition of atoms in the form of self-assembled metal-organic complexes [51] could be possible ways to make the lifetime of the daughter ion longer. Ideally, one should aim to tune the chemical environment so as to make the daughter ion chemically stable. Somehow or other, the lifetime of an ion formed after the beta decay process is determined by a range of environmental mechanisms and is a matter for the experiment to measure and manipulate. Here, we do not attempt to estimate it. Rather we investigate how the finite lifetime of the daughter ion influences the visibility of the C$\nu$B signal. Semi-empirically, one absorbs the effects of a finite lifetime into the Lorentzian broadening of the delta function representing the energy conservation law. For the spectral density function that amounts to an extra convolution [36].

$$A(E) \mapsto \int_{-\infty}^{\infty} A(\mathcal{E}) L(E-\mathcal{E}) d\mathcal{E}, \tag{60}$$

where $L(E)$ is the Lorentzian distribution function,

$$L(E) = \frac{\gamma/2}{\pi[E^2 + (\gamma/2)^2]}. \tag{61}$$

$\tau = \hbar/\gamma$ is the lifetime of the daughter ion. Thus the deformed spectral function should then be used in Eq. (18) to describe the observable beta spectrum. In particular, inserting Eq. (61)

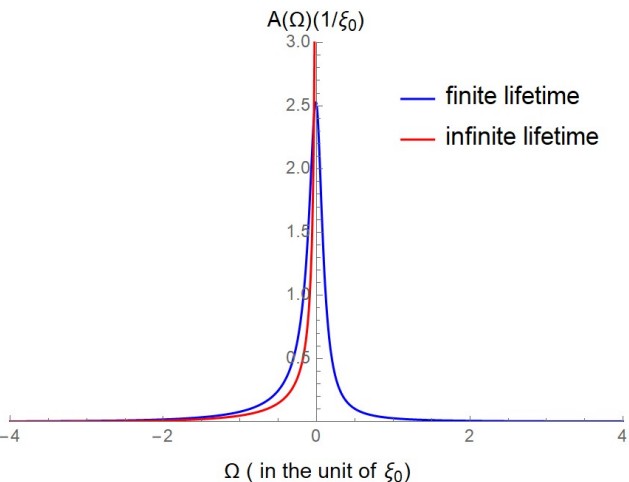

Figure 5: The spectral density function taking into account the influence of the finite lifetime of the daughter ion (blue curve) is compared with the spectral density function at infinite lifetime (red curve). As one can see, at a finite lifetime the the sharp Gamma distribution is broadened into a smooth asymmetric peak. In this plot we take $g = 0.125$, $\xi_0 = 1\,\mathrm{eV}$ and $\gamma = 0.2\,\mathrm{eV}$.

into Eq. (60), one obtains the shape of the spectral density function

$$A(E) = \mathrm{Im}\left[\frac{e^{\tilde{\Omega}}\Gamma(1-g,\tilde{\Omega})}{\pi\xi_0\tilde{\Omega}^{1-g}}\right], \tag{62}$$

where $\Gamma(x, y)$ is the incomplete gamma function and

$$\tilde{\Omega} = \frac{E - i\gamma/2}{\xi_0} = \Omega - \frac{i\gamma}{2\xi_0}. \tag{63}$$

To illustrate such a shape, we plotted $A$ as a function of the dimensionless eneregy $\Omega$ for $g = 0.125$, $\xi_0 = 1$ eV, and $\gamma = 0.2$ eV in Fig. 5. As one can see, at finite $\gamma$ the X-ray edge power law singularity gets deformed into an asymmetric bell-shaped distribution. The longer the lifetime, the closer the absorption peak is to the Gamma distribution (37).

While it is tempting to conclude that the main manifestation of a finite lifetime of the daughter ion is just some degree of broadening the neutrino absorption peak, which can be neglected for $\gamma \ll m_\nu$, on closer inspection the effect turns out to be a lot more detrimental and the conditions on $\gamma$ a lot more stringent. Moreover, the semi-empirical one-parameter model encapsulated in equation (61) turns out to be unsuitable for the quantitative analysis of the visibility of the C$\nu$B peak. Indeed, unlike the pure X-ray edge singularity (37), the semi-phenomenological distribution (62) has a long power-law tail on the right-hand side of the peak

$$A(E) \sim \frac{\gamma}{E^2}, \qquad E \gg \gamma. \tag{64}$$

At the same time, the beta-decay background near the emission edge $E_0 \approx Q$ has the parabolic shape

$$\frac{d\Gamma(E)}{dE_k} = \alpha\frac{N}{Q^3 T}(E_0 - E)^2, \qquad E < E_0, \tag{65}$$

which remains a good approximation up to $E_0 - E$ on the order of $Q$. In this expression $\alpha$ is a nucleus-specific numerical constant on the order of 1 and $T$ is the half-lifetime. Due to the rapid increase of the background intensity away from the termination point $E_0$, the

convolution (18) at energies slightly above the edge of the spectrum is completely dominated by the bulk of the beta spectrum and to a good accuracy can be expressed as

$$\frac{d\tilde{\Gamma}_{\text{BG}}}{dE_k} = C + o(E - E_0), \tag{66}$$

where

$$C = \frac{\gamma}{2\pi} \int_0^Q \frac{1}{(Q-\mathcal{E})^2} \frac{d\Gamma}{dE_k}(\mathcal{E}) d\mathcal{E} \sim \frac{\gamma N}{Q^2}. \tag{67}$$

In other words, the number of beta decay events as a function of the energy distance from the emission threshold increases so rapidly that it overwhelms the slow $1/E^2$ decay of the Lorentzian and thus creates a massive background in the region of the C$\nu$B peak. In reality, the fact that the main contribution to noise comes from the tail of the Lorentzian distribution invalidates the model (61) and the resulting estimate (67). To obtain an accurate result one would have to replace $L(E)$ with the exact ion's spectral function. Although such a spectral function is impossible to calculate in practice, one can still make an order of magnitude estimate of the background intensity in the C$\nu$B peak region based on the observation that the tail on the right-hand side of the resonant peak terminates at energies on the order of the ionisation energy of the recombined atom. More precisely, the termination point of the support of the ion's spectral function is defined by the process of direct transition of the beta-decayer into the neutral daughter atom and a positively charged quasihole on graphene's Fermi surface, which is the lowest energy out-state of the combined graphene - daughter atom system. In order to incorporate this physics into the semi-phenomenological model we introduce a truncated Lorentzian model in which $L(E)$ is given by Eq. (61) up to the first ionization energy $E_I$ of the daughter atom attached to the graphene and vanishes for $E > E_I$. Using such a truncation, the background will be approximately given by

$$\frac{d\tilde{\Gamma}_{\text{BG}}(E)}{dE_k} \sim \frac{\gamma}{2\pi} \int_{E-E_I}^{E_0} \frac{1}{(E-\mathcal{E})^2} \frac{d\Gamma(\mathcal{E})}{dE_k} d\mathcal{E}. \tag{68}$$

Such a function describes an $E_I$-wide tail of background events above the edge $E_0$ containing total events per unit time

$$\frac{dN_{\text{BG}}}{dt} \sim \frac{E_I^2 \gamma N}{TQ^3}. \tag{69}$$

The practical implications of the finite lifetime for the required amount of radioactive material are discussed in section 5.

## 5 Discussion

In this section, we have investigated how the results of the previous sections translate into the visibility of the C$\nu$B signal in the PTOLEMY experiment. From Eq. (18), we know that we can get the correct beta decay spectrum by performing a convolution between the spectral density function and the beta decay spectrum in the vacuum. Using Mathematica, we plotted the beta decay spectrum with, $m_{\text{lightest}} = 50$ meV. As can be seen from comparison of Fig. 6a and Fig. 6b, the shakeup of the electron system results in each cosmic neutrino background peak getting deformed into a strongly asymmetric shape peaking near the actual neutrino mass and having a long power-law tail stretching into the beta decay background. Note that the distortion of the beta-decay background is not as conspicuous as in the case of the C$\nu$B peak. The reason for that is that the distortion is a result of each individual beta-electron *losing*

some part of its energy in order to create a collective excitation of electrons in graphene. That is why the distortion of the C$\nu$B peak results in signals appearing at lower energies in the gap between the C$\nu$B signal and the beta-decay background. At the same time the distortion of the beta-decay background just leads to a certain amount of re-distribution of electrons within the beta-decay continuum, without affecting the endpoint. The latter effect is difficult to see by the unaided eye, especially on the logarithmic scale. Using the calculated spectrum, we obtained the visibility, that is the number of events in the non-overlapping region in the spectrum. For the calculation of visibility, we assume the amount of radioactive material equivalent to four capture events per year (about 100 g in the case of Tritium).

From our analysis, it follows that the visibility depends on four parameters, the lightest neutrino mass, the coupling constant $g$, the distance from the daughter ion to the graphene sheet, and the lifetime of the daughter ion. We investigate the PTOLEMY project's sensitivity to those parameters by making plots of the visibility in different parameter planes.

In Fig. 7 we show the dependence of the visibility on the lightest mass of neutrino and the coupling constant $g$. As shown in Fig. 7, the smaller the coupling constant and the larger lightest mass of the neutrino, the better the visibility. This result can be explained by the fact that a smaller coupling constant leads to a sharper distribution, which causes the beta decay spectrum less broadening. The effect of the distance between the daughter ion and graphene is that it affects the cutoff energy and therefore influences the shape of the distorted beta-emission peak. Smaller cutoff energy means less weight in the long tail of the gamma distribution, therefore fewer losses and better visibility. Indeed, as one can see from Fig. 9, visibility increases with the helium ion's height. Note that compared with the previous three parameters the influence of height on visibility is less pronounced, at least for a small enough coupling constant. This is explained by the fact that for small $g$ a lot of spectral weight is concentrated in the infrared region near the X-ray edge therefore the visibility is less sensitive to the ultraviolet cutoff.

To get some feel for the magnitude of the impact of the lifetime, we apply the estimate Eq. (69) to beta decay of Tritium neglecting for illustrative purposes the problems arising from recoil. A 100 g sample of Tritium contains $N = 2 \times 10^{25}$ atoms, and it is predicted to experience only 4 neutrino capture events per year. We take $E_I \sim 10$ eV and $Q \sim 10$ keV and $T \sim 10$ years and demand that $dN_{\mathrm{BG}}/dt$ be less than 1 event per year, which would roughly correspond to a $3\sigma$ detection confidence if 5 events are observed during the 1 year period. This results in the requirement $\gamma < 10^{-14}$ eV, that is a lifetime on the order of a few hundred milliseconds or longer. The condition on the lifetime of the daughter ion can in principle be relaxed if the energy window $(E_0, E_0 + \Delta E)$ containing the neutrino capture peak is known *a priori*. In that case, the number of background events inside that window is estimated as the right-hand side of Eq. (69) times $\Delta E/E_I$. For $\Delta E$ on the order of 100 meV that gives the lower bound on the ion's lifetime of about 1 ms. This estimate is further elaborated in our numerical calculation, illustrated in Fig: 8, of the effective amount of the radioactive material required in order to achieve the p-value of $5\sigma$ in a C$\nu$B detection experiment using the energy bin of 100 meV above the spontaneous beta-emission threshold of Tritium. We see that in order to achieve the signal-to-noise ratio ensuring $5\sigma$ confidence of detection of the $C\nu$B signal with a reasonable amount of the radioactive material the lifetime of the daughter ion needs to be greater than 100 $\mu$s. Such a stringent requirement for the lifetime of the daughter ion poses a separate challenge to the experiment. Detailed proposals as to how this challenge could be addressed will be discussed elsewhere.

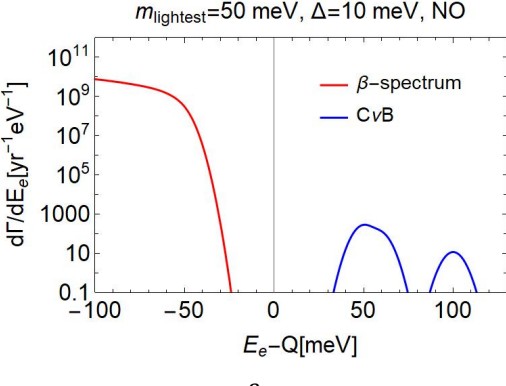
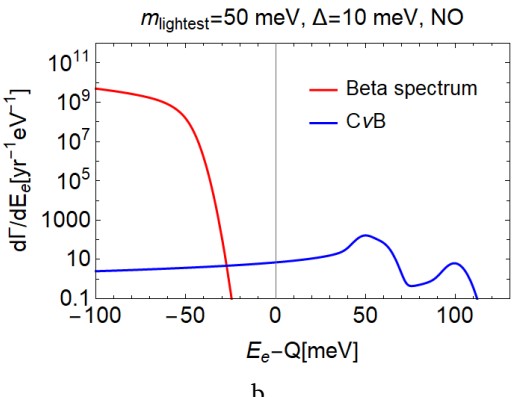

a                               b

Figure 6: Panel 6a shows the theoretical beta decay spectrum of mono-atomic tritium in the vacuum assuming the lightest neutrino mass of 50 meV The solid red curve represents the background beta decay. The solid blue curve represents the signal from the cosmic neutrino background. $d\Gamma/dE$ describes the possibility that the events happen at a given energy per year. The area under the curve represents the number of events per year. The area under the blue curve is only four, that is the calculation is done for the amount of Tritium that can capture at most four cosmic neutrinos per year. In the region where the beta decay background (red curve) overlaps with the C$\nu$B (blue curve), one cannot distinguish the cosmic neutrino signal from noise. Panel 6b describes the beta decay spectrum adjusted for the Fermi sea shakeup effect. We choose the coupling constant $g = 0.125$.

# 6 Summary and Outlook

In this work, we discussed how the shakeup of graphene's Fermi sea may influence the visibility of the cosmic neutrino signal in the PTOLEMY project. In the first section, we used Fermi's golden rule to find the beta decay spectrum adjusted for the Fermi sea shakup effect. We showed that it is the convolution between the spectral density function of a coulomb centre in graphene and the beta spectrum of the decaying isotope in the vacuum. Furthermore, we applied the linked cluster expansion technique, to obtain the spectral density function, which is the Gamma distribution function and is controlled by two parameters, the cutoff energy and the dimensionless coupling constant.

In the next section, we examined factors that influence the coupling constant and the cutoff energy, and we also considered the influence of the daughter ion's lifetime on the spectral density function. We found that the distance between the ion and graphene dictates the natural distance-dependent energy cutoff scale. Also, the effects of the disorder can be neglected if the mean free time of an electron in graphene exceeds the resolution time of the experiment, which is about $10^{-13}\,s$. However, the dynamic screening effects of the intrinsic graphene significantly increase the coupling constant, which is detrimental to the visibility of the PTOLEMY project. Fortunately, if the external dielectric constant of the substrate is large enough, the dynamic screening effects can be suppressed. We further established that the lifetime of the daughter ion may have a hugely detrimental impact on the discernibility of C$\nu$B signal. To illustrate this point we show that for the decay of a reasonable amount of Tritium, the lifetime of the Helium ion would need to be at least $100\,\mu s$ or longer.

Overall, we have three main conclusions.

(1) Our preliminary analysis indicates that despite large energy scales associated with the shakeup of the Fermi sea, the visibility can still be protected by the X-ray edge singularity and screening effects. Even though we use the parameters of Tritium for numerical illustration,

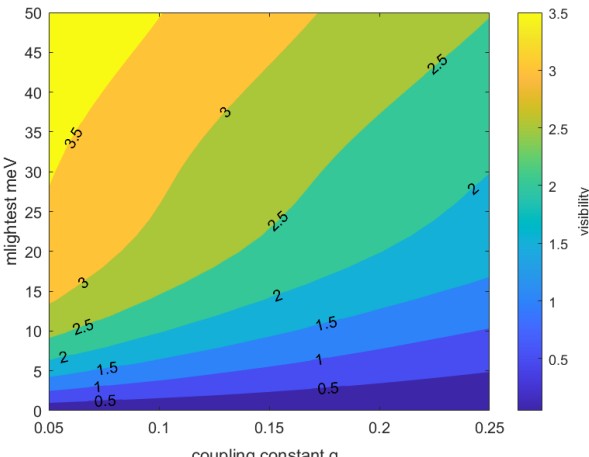

Figure 7: Contour plot of visibility as a function of the lightest mass of neutrino and the coupling constant. The cut-off energy is assumed to be 1 eV. The maximum possible number of C$\nu$B events is 4.

our results are universal and directly adaptable to any other atom.

(2) The visibility of the C$\nu$B signal can be improved in the following ways:

1) use a substrate with a high dielectric constant;

2) deposit the radioactive atoms in such a way as to maximize their distance from the graphene sheet, possibly with the help of a dielectric spacer

(3) Not all effects are included in our analysis, so experiments should be conducted to test the validity range of our theory. Also, the parameters such as the coupling constant $g$ and the lifetime of the daughter ion can only be reliably determined in an experiment.

We finally remark that although we have considered some important solid state effects that can affect the spectral density function, other effects require further study. Those include phonon emission, inhomogeneous broadening in a disordered sample, Coulomb interaction of the beta-electron with electrons in graphene and others. While such effects are not necessarily important in the traditional X-ray emission experiment, the extraordinary energy resolution required by the PTOLEMY experiment puts unusually stringent constraints on the admissible rate and magnitude of disruption due to various solid state processes. We hope that further research will improve our understanding of such effects and their mitigation.

## Acknowledgments

We wish to acknowledge Dr. Boyarksy for providing us with the data. We also appreciate the discussion with Yevheniia Cheipesh. The first author especially wants to express his gratitude to his wife Yiping Deng for taking care of him when writing this paper.

## A   Appendix 1

The details of obtaining the natural cutoff energy are presented here.

Substituting Eq. (39) into the Eq. (27), one can find that

$$F_2'(t) = -g \int_0^{\xi_0} \frac{(1-e^{-iut})du}{u} - \frac{\pi g}{2} \int_0^{\xi_0} du \frac{(1-e^{-iut})}{u}\left[L_1\left(\frac{2du}{v_F}\right) - I_1\left(\frac{2du}{v_F}\right)\right]. \quad \text{(A.1)}$$

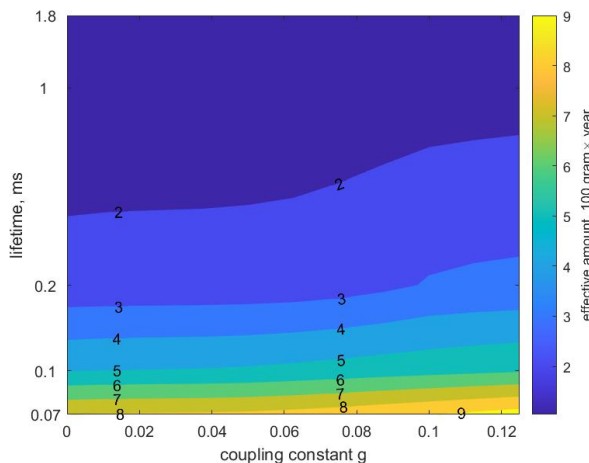

Figure 8: Contour plot of the effective amount of Tritium required in order to achieve the p-value of $5\sigma$ or better as a function of the coupling constant and the lifetime. The unit of the effective amount of tritium is 100 gram × year, which means one can achieve the same necessary number of events by either increasing the amount of Tritium or increasing the duration of the experiment. The plot is calculated within the truncated Lorentzian model assuming the ionization energy $E_I = 10$ eV.

The first integration is the original term, labelled as $F_2^0(t)$ and the second integration is the correction term, labelled as $F_2'(t)$. Directly evaluating the correction term is difficult, but we can express it as a double integral.

$$F_2'(t) = g \int_0^{2d\xi_0/v_F} \int_0^1 e^{-zx}\sqrt{1-x^2}(1-e^{-izv_Ft/2d})dxdz = P(\xi_0) + N(\xi_0, t), \qquad (A.2)$$

where

$$P(\xi_0) = g \int_0^1 dx \sqrt{1-x^2}\left(\frac{1-e^{-2dx\xi_0/v_F}}{x}\right), \qquad (A.3)$$

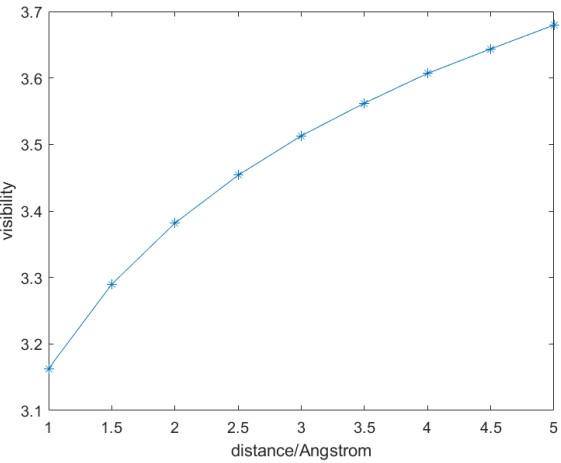

Figure 9: Visibility at different heights with coupling constant $g = 0.1$, and $m_{\text{lightest}} = 50$ meV. The curves with different coupling constants and the neutrinos' lightest mass share a similar feature.

and

$$N(\xi_0, t) = g \int_0^1 dx \sqrt{1-x^2} \left( \frac{-1 + \exp(-it\xi_0 - 2dx\xi_0/v_F)}{x + iv_F t/2d} \right),$$ (A.4)

with $z = 2du/v_F$. From the calculation above, we can find that the correction term can be split into two terms. The first part is a constant independent of time t, and the second part is a time-dependent function. Hence, only the function $N(\xi_0, t)$ is significant to our calculation. Obviously, if $t = 0$, it cancels with the first term $P(\xi_0)$. However, when t approaches infinity, the correction term $F_2'(t)$ should equal to $P(\xi_0)$. We need to make some approximations to evaluate the function $N(\xi_0, t)$.

At a large time limit, roughly, we can use 1 to replace the ugly square root $\sqrt{1-x^2}$ since most contributions to the integration in $P(\xi_0)$ and $N(\xi_0, t)$ come from the vicinity of x=0. Then, we can write the correction term $F_2'(t)$ in a simple way.

$$\begin{aligned}
F_2'(t) &= g \int_0^1 \left( \frac{1 - e^{-2dx\xi_0/v_F}}{x} \right) dx + g \int_0^1 dx \left( \frac{-1 + \exp(-it\xi_0 - 2dx\xi_0/v_F)}{x + iv_F t/2d} \right) \\
&= g \int_0^1 \left( \frac{1 - e^{-2dx\xi_0/v_F}}{x} \right) dx - g \int_{iv_F t/2d}^{1+iv_F t/2d} \left( \frac{1 - \exp(-2dy\xi_0/v_F)}{y} \right) dy \\
&= g \int_0^1 \left( \frac{1 - e^{-2dx\xi_0/v_F}}{x} \right) dx + g \int_0^{iv_F t/2d} \left( \frac{1 - \exp(-2dy\xi_0/v_F)}{y} \right) dy \\
&\quad - g \int_0^{1+iv_F t/2d} \left( \frac{1 - \exp(-2dy\xi_0/v_F)}{y} \right) dy.
\end{aligned}$$ (A.5)

The factor $1 - \exp(-2dy\xi_0/v_F)$ behaves as $2dy\xi_0/v_F$ in the limit $u \to 0$. We can use $\frac{v_F}{2d\xi_0}$ to replace the lower limit of the integration, since $\frac{v_F}{2d\xi_0} \ll 1$. At very large time t, we omit the factor $\exp(-2dy\xi_0/v_F)$. Hence, the correction term $F_2'(t)$ can be approximately expressed as a very neat form.

$$\begin{aligned}
F_2'(t) &\approx g \int_{v_F/(2d\xi_0)}^1 \frac{dx}{x} + g \int_{v_F/(2d\xi_0)}^{iv_F t/2d} \frac{dy}{y} - g \int_{v_F/(2d\xi_0)}^{1+iv_F t/2d} \frac{dy}{y} \\
&\approx g \ln \left( 1 + \frac{2d\xi_0}{v_F} \right) + g \ln(i\xi_0 t + 1) - g \ln \left( i\xi_0 t + \frac{2d\xi_0}{v_F} + 1 \right).
\end{aligned}$$ (A.6)

Substituting Eq. (A.6) into equation (A.1), and using equation (25) and equation ((37)), one can obtain the expression of the spectral density function of graphene.

$$\begin{aligned}
A(\Omega') &= \int_{-\infty}^{\infty} \frac{dt}{2\pi} e^{-iEt} \exp(F_2^0(t) + F_2'(t) \\
&= \Theta(-\Omega') \frac{\exp(\Omega')}{\Gamma(g)\xi'(-\Omega')^{(1-g)}},
\end{aligned}$$ (A.7)

where $\xi' = \frac{\xi_0}{1+2d\xi_0/v_F}$, and $\Omega' = (E + E_i)/\xi'$.

The cut-off energy $\xi_0$ is entirely arbitrary, but we can get a natural cut-off energy $\xi$ when we extend $\xi_0$ to infinity

$$\xi = \lim_{\xi_0 \to \infty} \frac{\xi_0}{1 + 2d\xi_0/v_F} = \frac{\hbar v_F}{2d}.$$ (A.8)

We get nearly the same result as the previous one, Eq. (37), only with different cutoff energy $\xi$.

SciPost Phys. **17**, 022 (2024)

# B  Appendix 2

For intrinsic graphene, the time-dependent potential is

$$V(q,t) = \frac{V(q)\theta(t)}{\epsilon(q,0)} + V(q)\theta(t)T(v_F q t). \tag{B.1}$$

$T(v_F q t)$ is a one-variable fast-decay function that only depends $v_F q t$.

$$T(v_F q t) = -\frac{2}{\pi}\int_1^\infty \frac{a\cos(\omega v_F q t)\sqrt{\omega^2-1}}{\omega(\omega^2-1+a^2)}d\omega, \tag{B.2}$$

where $a = (\epsilon(q,0)-1) = \frac{\pi e^2}{2\hbar\kappa v_F}$. When $t << 1$, $T(v_F q t) \approx \frac{1}{\epsilon(q,0)} - 1$, and when $t >> 1$, $T(v_F q t) \approx 0$.

From the general expression of the $F_2(t)$, Eq. (27), we can find that

$$\begin{aligned}
F_2(t) = &\frac{1}{2}\int_0^t dt_1 \int_0^t dt_2 \int_0^\infty e^{-iu(t_1-t_2)}\frac{du}{\pi V} \\
&\times \sum_q \left[\frac{1}{\epsilon^2(q,0)} + \frac{T(v_F q t_1)}{\epsilon(q,0)} + \frac{T(v_F q t_2)}{\epsilon(q,0)} + T(v_F q t_1)T(v_F q t_2)\right]|V(q)|^2\,\Lambda(q,u).
\end{aligned} \tag{B.3}$$

When $t \ll 1$, then we can find $F_2(t) \approx 0$. It is trivial and has no contribution to the spectral density function. Therefore, the most significant part of the X-ray edge is in the long-time domain, and the function $T(v_F q t)$ decays very fast, so $T(v_F q t_1)T(v_F q t_2)$ is negligible compared to other terms in the bracket. Therefore, one can get

$$\begin{aligned}
F_2(t) \approx F_2^0(t) &- \frac{gu\epsilon}{2}\int_0^t dt_1 \int_0^t dt_2 \int_0^\infty e^{-iu(t_1-t_2)}du \times \int_1^\infty \frac{aH_{-1}(u\omega t_1)\sqrt{\omega^2-1}}{\omega(\omega^2-1+a^2)}d\omega \\
&- \frac{gu\epsilon}{2}\int_0^t dt_1 \int_0^t dt_2 \int_0^\infty e^{-iu(t_1-t_2)}du \times \int_1^\infty \frac{aH_{-1}(u\omega t_2)\sqrt{\omega^2-1}}{\omega(\omega^2-1+a^2)}d\omega,
\end{aligned} \tag{B.4}$$

where $F_2^0(t)$ is the original term in Eq.(27) without considering dynamic screen effect. In the long time limit, we can extend t into infinity, and then one can get

$$\begin{aligned}
F_2(t) &\approx F_2^0(t) + \int_{i/t}^{\xi_0} \frac{2g\epsilon}{\pi u}du \int_1^\infty \frac{a\log(\omega+\sqrt{\omega^2-1})}{\omega^2(\omega^2-1+a^2)}d\omega \\
&\approx \log(1+g_1 i\xi t),
\end{aligned} \tag{B.5}$$

with

$$g_1 = g + \int_1^\infty \frac{2g\epsilon}{\pi}\frac{a\log(\omega+\sqrt{\omega^2-1})}{\omega^2(\omega^2-1+a^2)}d\omega. \tag{B.6}$$

Therefore, the spectral density function has the same form as the previous result in Eq. (37), but with a different coupling constant.

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
