# Peer review of "Importance of the X-ray edge singularity for the detection of relic neutrinos in the PTOLEMY project"

_SciPost Physics, doi:SciPost Phys. 17, 022 (2024)_

## Round 1 · Referee Report · Anonymous (Referee 1) · 2024-3-9

Strengths

1- the manuscript addresses a topic of great relevance. To dig deeper into the problem of condensed matter effects in the PTOLEMY project is important, and I believe this paper performs a number of step forward.

2- the main intuitive ideas behind the physical effects the authors are describing are presented in a clear way.

3- after pointing out the difficulties related to the detection of relic neutrinos, the authors give semi-concrete recipes on how to mitigate them. This can be of great help to try and identify possible directions to evade the issue.

Weaknesses

1- in more than one point the authors are a bit cavalier about notation, they miss to define quantities and there is a certain number of typos here and there. This makes sometimes reading the paper hard.

2- the discussion about the relevant time scale is a bit vague and, to me, unclear. I think the authors should make an effort to explain it more properly. It is my opinion that this represent probably one of the most important aspects to pin down: what is the relevant time scale that allows one to tell that certain processes are relevant and some others are not? (see "requested changes" for more details.)

Report

In general, I believe this manuscript to be extremely valuable.
It is definitely worth publication, but only after my previous points will be properly addressed.

Requested changes

1- at the beginning of section 2B it is mentioned that this work neglects the recoil of the daughter nucleus. It is unclear to me whether this is done for the sake of simplicity or not. Would the recoil of the final He+ introduce further distortion in the final electronic spectrum?

2- in Eq. (8) the authors introduce the Hamiltonian H_{D-G}. It seems to me that this is the same as the Hamiltonian H_g introduced in Eq. (5). Is this correct? If yes, I would make the notation uniform.

3- the states | ... >_h and | ... >_t in Eq. (16) are not defined. Do they stand for "helium" and "tritium"? It should be explained.

4- is the "i" appearing in the numerator of Eq. (52) an imaginary unit, or a mis-typed index?

5- in Eq. (50) the authors present an estimate of the density of tritium atoms that would allow to neglect disorder effects. I think it would be very useful to compare it with standard graphene density or, even better, to convert to to a mass density. This would allow a more immediate experimental comparison.

6- the various figures are pretty ill-placed. It would be much clearer if they were placed in the same page where they are first mentioned. This way the reader does not have to skim more of the manuscript before getting to them.

7- the only major physical point that confuses me is the following. When discussing the relevance of effects like the hole production in Eq. (7) or the interaction between electrons and impurities (sec. 4B), the authors compare the typical time scale associated to this processes to the inverse energy resolution of the experiment, resorting to the time-energy uncertainty principle. However, I naïvely would have expected the relevant time scale to compare it with to be the formation time of the beta-electron. All processes happening after that are essentially different way of redistributing the energy inside the graphene system which, however, should then be decoupled from the outgoing electron. I believe this point to be crucial, and I would to understand if the authors have a quantitative way of determining the relevant time scale to be used as benchmark. As is well known, the time-energy uncertainty principle is always a bit hard to put on firm grounds.

  • validity: high
  • significance: high
  • originality: high
  • clarity: good
  • formatting: good
  • grammar: perfect

Author:  Zhiyang Tan  on 2024-05-08  [id 4480]

(in reply to Report 1 on 2024-03-09)
Category:
answer to question

Response to the Referee report Zhiyang Tan and Vadim Cheianov

We would like to thank the Referee for the thorough review of our manuscript titled ”Importance of the X ray edge singularity for the detection of relic neutrinos in the PTOLEMY project.” We greatly appreciate the time and effort you have invested in providing constructive feedback to improve the quality of our work. We have carefully considered each of the Referee’s points and we address them below to the best of our ability:

1) The recoil of the Daughter Nucleus (Section 2B): We do not include the impact of the recoil in our model for the sake of simplicity. The impact of the recoil has been addressed in Ref.[17] of our manuscript, and the ways to mitigate the effect of recoil have been addressed in further literature (Refs. [18-24] in the manuscript). We assume that one of the proposed mitigating solutions has been implemented and therefore we focus on the next important harmful effect, which is the shakeup of the Fermi Sea in graphene. If necessary, incorporating the zero-point motion of the nucleus in our model is straightforward - one simply has to compute the convolution of the spectrum obtained in the present manuscript with the Gaussian distribution due to the zero-point motion as described in e.g. Ref. [17].

2) Notation Uniformity (Eq. 8): The Hamiltonian $H_{D−G}$ introduced in Eq. (8) is not equivalent to the Hamiltonian Hg introduced in Eq. (5). The former is the interaction between the ion and the graphene, while the latter is the full Hamiltonian of the graphene system, which also includes the kinetic energy of the electrons of graphene. The definition of $H_{D−G}$ is in Eq. (11) and the definition of $H_g$ is in Eq. (21).

3) Definition of States (Eq. 16): We appreciate the Referee’s observation regarding the undefined states $|... \rangle_{h} $ and $|... \rangle_{t}$ in Eq. (16). To keep the uniformity and consistency, we have changed them to $|... \rangle_{D}$ and $|... \rangle_{M}$. $|0\rangle_{M, D}$ denotes a state where the isotope (mother or daughter) is absent, and $|1\rangle_{M, D} $denotes a state where the isotope is present in the state of zero kinetic energy. We have also addressed them in the main text.

4) Presence of ”i” in Eq. (52): The ”i” appearing in the numerator of Eq. (52) is not a misprint and it indeed represents the imaginary unit. The expression is a way to express the Fourier transform of a sudden external potential, which is also addressed in Ref [52] of our manuscript.

5) Estimation of the Density of Tritium (Eq. 50): We thank the Referee for suggesting a comparison of the estimated Tritium density with the standard graphene density or conversion to mass density for easier experimental comparison. We agree that this would enhance the accessibility of our results and we have included such comparisons in the revised manuscript.

6) Placement of Figures: We acknowledge the Referee’s feedback regarding the placement of figures and agree that it would improve clarity if they were placed on the same page where they are first mentioned. We have done our best to reorganize the figures accordingly, however, some of the figures still appear on a page, which is different from the page of the first mention. Such a placement is not our fault, but it is due to the internal formatting algorithms of LaTeX. 7) Relevant Time Scale for Physical Processes: The referee is, in a sense, correct in saying that all processes happening after beta decay are essentially different ways of redistributing the energy inside the graphene system. However, the statement that these processes are decoupled from the outgoing electron contains a subtle ambiguity, which may be the source of the confusion. We shall illustrate this point with the example of the neutrino capture process, although the arguments also apply to beta-decay.

Firstly, let us remark that the formation time of the beta electron is extremely short (on the order of $10^{−16 }$seconds) and the associated energy scale runs into $KeV$. Clearly, this energy scale has no relevance to the problem of the broadening of the beta-emission line in the neutrino capture process. In fact, the limit in which the formation of the beta electron is treated as instantaneous is mathematically sound and is a good approximation for the treatment of soft excitations created in the process.

Now, following the capture, which we treat as an instantaneous process, the combined graphene + daughter ion + beta-electron system emerges in a quantum superposition of states $|f\rangle= \sum_{k,λ} C_{k,λ}|k\rangle_e|λ\rangle$, where $|λ\rangle$ are Eigenstates of the Hamiltonian of the (graphene + daughter ion) system and $|k\rangle_e $are the eigenstates of the kinetic energy of the beta-electron. The states in the superposition have to obey the energy conservation law $(\hbar^2k^2/2m_e)+E_λ = const$ where the constant on the right-hand side is dictated by the mass defect difference. Due to this constraint, the tensor $C_{k,λ}$ is fundamentally inseparable, and the final state is an entangled state between the outgoing electron and the graphene system in which the kinetic energy of the beta electron is indefinite. Therefore despite being decoupled from graphene and the daughter ion in the dynamical sense, the beta electron remains entangled with this system until the projective measurement of its energy is performed in the calorimeter. This entanglement necessitates the total transition probability to be a convolution of the transition probability function of beta decay and the graphene spectral density function. The influence of the energy redistribution between the graphene system and the daughter ion on the energy spectrum is encapsulated within the spectral density function, defined by Eq. (20).

On a more general note, the effect of the finite lifetime of the products of a reaction on the energy conservation law in that reaction is well-known and described in the literature. The bottom line is that processes leading to the collisional decay of the products of a reaction, even if they happen long after the reaction itself took place, have an effect on the available phase space of the reaction due to the broadening of the ”on-shell” delta-function into a more generous spectral density function, typically a Lorentzian. See, e.g.

L. P. Kadanoff and G. Baym, Quantum Statistical Mechanics (Benjamin, New York, 1962/ Perseus Books, Cambridge, MA, 1989).

J. R. Barker, Quantum Theory of high-field transport in semiconductors. Journal of Physics C 6, 2663 (1973)

Once again, we thank you for your valuable feedback, which will undoubtedly strengthen the quality and clarity of our manuscript. We will make the necessary revisions as suggested and look forward to resubmitting the improved version for your consideration.

---

## Round 1 · Referee Report · Anonymous (Referee 2) · 2024-5-19

Report

This paper touches on the very important problem of how solid state effects and quantum broadening may impact CMB neutrino detection. The current paper constitutes a notable study in whether this broadening may adversely impact the PTOLEMY experiment. In previous work, the Fermi shakeup effect was introduced and examined for a single Tritium ion on a graphene surface in an idealized setting. The current article goes far beyond earlier work. I applaud the very careful analysis of the authors and its presentation. At this end, it was illuminating to learn of the core hole recombination and of the (not so severe) outcome of the detailed calculations for the Fermi shakeup and how the latter may, in principle, be mitigated.

There are no real required changes at this end apart from that of correcting the very few minor typos (e.g., "niclei").

I very strongly recommend the publication of this manuscript.

Recommendation

Publish (surpasses expectations and criteria for this Journal; among top 10%)

---

## Round 2 · Referee Report · Anonymous (Referee 1) · 2024-5-27

Strengths

1- improved clarity of the manuscript. 2- thorough discussion of the consequences of entanglement between beta-electron and the rest of the system

Weaknesses

1- too technical in some parts

Report

In the second version of their manuscript, the authors have made an effort to accommodate my requests and clarify my doubts.
I think the paper has now improved in clarity and I have personally understood better the physics behind.
I am glad to recommend it for publication.

Recommendation

Publish (easily meets expectations and criteria for this Journal; among top 50%)

---

## Editorial Decision

published